# Reversal of T Cell Exhaustion in Chronic HCV Infection

**DOI:** 10.3390/v12080799

**Published:** 2020-07-25

**Authors:** Sylwia Osuch, Karin J. Metzner, Kamila Caraballo Cortés

**Affiliations:** 1Department of Immunopathology of Infectious and Parasitic Diseases, Medical University of Warsaw, 02-091 Warsaw, Poland; sosuch@wum.edu.pl; 2Division of Infectious Diseases and Hospital Epidemiology, University Hospital Zurich, University of Zurich, CH-8091 Zurich, Switzerland; karin.metzner@usz.ch; 3Institute of Medical Virology, University of Zurich, CH-8057 Zurich, Switzerland

**Keywords:** chronic HCV infection, T cell exhaustion, inhibitory receptors, direct acting antivirals

## Abstract

The long-term consequences of T cell responses’ impairment in chronic HCV infection are not entirely characterized, although they may be essential in the context of the clinical course of infection, re-infection, treatment-mediated viral clearance and vaccine design. Furthermore, it is unclear whether a complete reinvigoration of HCV-specific T cell response may be feasible. In most studies, attempting to reverse the effects of compromised immune response quality by specific blockades of negative immune regulators, a restoration of functional competence of HCV-specific T cells was shown. This implies that HCV-induced immune dysfunction may be reversible. The advent of highly successful, direct-acting antiviral treatment (DAA) for chronic HCV infection instigated investigation whether the treatment-driven elimination of viral antigens restores T cell function. Most of studies demonstrated that DAA treatment may result in at least partial restoration of T cell immune function. They also suggest that a complete restoration comparable to that seen after spontaneous viral clearance may not be attained, pointing out that long-term antigenic stimulation imprints an irreversible change on the T cell compartment. Understanding the mechanisms of HCV-induced immune dysfunction and barriers to immune restoration following viral clearance is of utmost importance to diminish the possible long-term consequences of chronic HCV infection.

## 1. Introduction

WHO’s Global Hepatitis Report estimates that about 71 million people are infected with hepatitis C virus (HCV) worldwide [1]. HCV is a blood-borne virus which is mainly transmitted via parenteral exposure as a consequence of intravenous drug use, or through reuse of injection needles or syringes [2]. The risk of parenteral infection is also increased among health care employees with frequent exposure to blood. HCV can also be transmitted sexually through permucosal exposure, especially in individuals with multiple sex partners, or in HIV-infected men who have sex with men (MSM) [2,3]. Furthermore, mother-to-child vertical transmission occurs in 2–8% of HCV-infected mothers [4]. About 80% of HCV-infected individuals do not exhibit any symptoms at the initial stage of infection, while the remaining 20% of acute cases develop mildly symptomatic infection. The virus is cleared spontaneously in 15–45% of individuals with acute infection, whereas the remaining 55–85% of cases develops chronic hepatitis C [5].

Clinical observations show that approximately 20% of chronically-infected patients develop advanced fibrosis and cirrhosis [6], which may lead to hepatocellular carcinoma (HCC), causative of approximately 399,000 of deaths per year [1,2]. HCV-induced HCC usually develops within 20–40 years of infection and is observed in about 1–7% of infected patients with liver cirrhosis per year [7]. 

Currently, there is no vaccine available preventing HCV infection, but advances in diagnostic procedures and direct-acting antiviral (DAA) treatment resulted in substantial improvement in clinical care of individuals with hepatitis C, which is crucial in preventing HCV-related morbidity and mortality. Nevertheless, despite highly effective therapeutic options, the risk of HCC development in cirrhotic patients persists even after successful treatment [8].

The extraordinary genetic heterogeneity of HCV is a result of fast replication and high error rate of viral RNA-dependent RNA polymerase and is manifested by the *quasispecies* phenomenon, the concomitant presence of closely related genetic variants within an infected host, largely facilitating the adaptive dynamics of the virus [9]. HCV genetic heterogeneity is a major mechanism of immune system evasion, because of the increased probability of positive selection of escape variants in the immune pressure of the host [10]. The occurrence of mutations within the viral T cell epitopes was associated with diminished recognition by virus-specific T cells [11]. Viral escape occurs early during acute infection, indicating that it contributes to HCV persistence [12], but is also observed in approximately 50% to 70% of viral epitopes targeted by virus-specific CD8^+^ T cell in chronic infection [12,13].

## 2. T Cell Exhaustion in HCV Infection

Adaptive immune responses play a critical role in the clinical course of infection with HCV [14,15]. HCV elimination coincides with strong and sustained multi-specific CD4^+^ and CD8^+^ T cell immunity which remains detectable after the spontaneous resolution of infection [15]. However, the quality of this response is substantially deteriorated once chronic infection is established [16]. Both CD4^+^ and CD8^+^ HCV-specific T cells are commonly present in liver tissue and in peripheral blood, however, in most patients, these cells are unable to clear the infection and do not prevent re-infection with HCV [14,15,17]. The underlying immune impairment phenomenon has been termed T cell exhaustion, defined as weak antigen-specific T cell responses, manifested as the deterioration in antiviral effector functions of antigen-specific T cells, such as a decline in effector cytokines’ production, the decreased capability to eliminate infected cells and impaired proliferation after antigen exposure in vitro [18,19]. The consequence of this phenomenon is loss of control over the ongoing infection, and emerging data suggest that exhaustion is a crucial factor determining viral persistence [20,21,22,23]. T cell exhaustion is not uniquely observed in HCV infection, but also in other chronic viral infections, particularly with lymphocytic choriomeningitis virus (LCMV), human immunodeficiency virus (HIV) or hepatitis B virus (HBV), as well as in tumors [20,24,25,26,27]. 

Although most findings are based on the LCMV mouse model, the pathway of T cell exhaustion seems to be universal. The decline in T cell effector functions is sequential and hierarchical, being initiated by the loss of interleukin (IL)-2 expression, followed by the decreased expression of tumor necrosis factor (TNF) and ultimately interferon (IFN)-γ, β-chemokines, as well as impaired cytotoxicity [28,29]. Moreover, exhausted CD8^+^ T cells downregulate the expression of IL-7 and IL-15 receptors, which physiologically sustain the proliferation and survival of memory T cells [30,31,32]. Despite substantial functional impairment, exhausted T cells may continue to express proteins associated with effector function [27]. It is believed that T cell exhaustion has evolved as a host-driven mechanism to limit the severity of the immune response and protect from immunopathology [33]. 

T cell exhaustion is mediated by continuous antigen stimulation, progresses along the time of infection, and is accompanied by transcriptional, translational, metabolic, nucleosomal and epigenetic changes [34,35,36,37,38]. In consequence, exhausted T cells display a characteristic phenotypic and functional pattern distinct from effector and memory T cells, pointing out that exhaustion represents a separate branch of CD8+ T cell differentiation [39,40,41]. 

On a phenotypic level, T cell exhaustion during chronic infection is manifested as upregulation of inhibitory receptor (iR) protein molecules, which deliver negative signals precluding cell activation after antigen recognition and downregulate the functional and proliferative potential of the responding cells [37,40,42]. In acute infection, iRs function to limit immune responses, but are downregulated when the pathogen is cleared. It has been demonstrated that iRs negatively affect T cell function and activation at several levels: (i) through competition with co-stimulatory receptors for shared ligands; (ii) by interfering with signals from co-stimulatory receptors or TCR; (iii) by the upregulation of genes involved in T cell dysfunction [43,44]. IRs, which have been linked to T cell exhaustion, include but are not limited to programmed cell death-1 (PD-1/*CD279*), cytotoxic T cell antigen 4 (CTLA-4/CD152), B- and T-lymphocyte attenuator (BTLA/CD272), CD160, CD200, NK cell type I receptor protein (2B4/CD244), lymphocyte-activation gene 3 (LAG-3), T cell immunoreceptor with Ig and ITIM domains (TIGIT), Ig superfamily-related receptors (GP49/CD85k) and T cell immunoglobulin and mucin-domain containing-3 (Tim-3) (reviewed in [40,43,44]). Furthermore, molecular iR pathways seem to be mechanistically related, and it is the co-expression of multiple iRs defining the state of exhaustion rather than their mono-expression, which was confirmed by the observation that the simultaneous blockade of several iRs results in better effects on the reversal of exhaustion [40].

The underlying changes in transcriptional program which drive the phenotypic and functional signature of exhausted CD8^+^ T cells are complex, specific and sequential [45]. An additional factor complicating this landscape is the heterogeneity of exhausted CD8^+^ T cell populations [46,47,48,49]. Although a number of transcription factors have been attributed to the pathogenesis of T cell exhaustion (e.g., B lymphocyte-induced maturation protein-1 (Blimp-1), basic leucine zipper ATF-Like transcription factor (Batf), eomesodermin (Eomes), and nuclear factor of activated T cells (NFAT) as well as T-box protein expressed in T cells (T-bet/TBX21) [48,50,51,52,53], recent discoveries pointed to the paramount role of T cell factor family member TCF-1 (Tcf7), as well as HMG-box transcription factor TOX [23,54,55]. TCF-1 is a key transcription factor for the ‘‘stem-like’’, progenitor exhausted PD-1^+^TCF-1^+^ CD8^+^ T cell population, which sustains CD8^+^ T cell responses during chronic viral infection and gives rise to PD-1^+^Tim3^+^ TCF-1^−^ exhausted and terminally differentiated cytotoxic cells [55,56]. Similarly, TOX has been identified as a critical transcriptional and epigenetic coordinator of exhausted CD8^+^ T cell programming, being robustly expressed in exhausted T cells, but transiently and at low levels during acute viral infection. TOX is necessary and sufficient to induce major features of exhausted T cells, including transcriptional changes, the expression of iRs and decreased function [23,54]. The establishment and maintenance of exhausted T cells depends on TOX, TCF-1, Eomes and T-bet as essential components regulating their development and balance [36,48,55,57]. Beltra et al. delineated a four-stage developmental exhausted CD8^+^ T cell hierarchy driven by transcription factor cascade conversion from progenitor 1 quiescent and resident TCF-1^hi^TOX^hi^ to progenitor 2 proliferative circulating TCF-1^int^TOX^hi^ to intermediate circulating mildly cytotoxic TCF-1^neg^T-bet^hi^TOX^int^ and, finally, to terminally exhausted resident TCF-1^neg^T-bet^lo^TOX^hi^Eomes^hi^ cells [36]. 

In contrast to CD8^+^ cells, CD4^+^ T cell exhaustion is still poorly understood, which is mainly due to the lower ex vivo frequency of peripheral blood virus-specific CD4^+^ T cells during chronic infection as well as technical limitations, particularly in generating MHC class II/antigen complexes by recombinant methods, and lower affinity binding of CD4^+^ TCR to cognate epitope on MHC class II molecule [58,59]. To date, it has been shown that exhausted CD4^+^ T cells lose the ability to secrete TNF-α, IFN-γ and IL-2, but increase production of IL-10 and IL-21 [60,61]. Furthermore, some data suggest that virus-specific CD4^+^ T cells lose effector function sooner than CD8^+^ T cells [62]. 

Although phenotypically similar, exhausted CD4^+^ and CD8^+^ T cells display certain qualitative differences. In particular, CTLA-4 and CD200 expression, as well as an increase in KAROS family zinc finger 2 (Ikzf2) (Helios) and Kruppel-like factor 4 (Klf4), are more specific to CD4^+^ T cells [63].

HCV infection represents an extraordinary opportunity to study the pathogenesis of T cell exhaustion in humans, since the spontaneous and complete resolution of infection is uniquely feasible among human chronic viral infections, even after exposure to a strain which previously established chronic infection in other subjects [64]. Furthermore, being the only chronic viral infection in both humans and chimpanzees which can be cured by a highly specific, small molecule-based DAA treatment, the effect of the removal of constant stimulation with viral antigens, the underlying cause of immune exhaustion, can be genuinely studied [49,65]. The use of magnetic bead enrichment of HCV-specific T cells, as well as recent advances in next-generation sequencing technologies, facilitated studies of these extremely rare populations [16]. Research on the dysregulation of transcriptional, metabolic, nucleosomal, and immune processes in HCV-specific CD8^+^ T cells preceding the overt establishment of T cell exhaustion in this infection is currently underway. Recently, the transcriptomic profile of HCV-specific CD4^+^ and CD8^+^ T cells during acute resolving vs. chronic infection has been delineated. Similar to the LCMV mouse model, it was found that TOX is induced by high antigen stimulation of the T cell receptor during chronic HCV infection in humans and correlates with the exhausted phenotype, in particular with PD-1 expression levels [23]. Furthermore, TOX was detectable in chronic but not in spontaneously resolved HCV infection or influenza-specific memory T cells, pointing out that TOX is crucial for direction of adaptive T cell responses toward exhaustion. Similarly, progression to chronic HCV infection was characterized by higher expression of strategic regulators of T cell immune function, proliferation, and survival, i.e., TBK1, SIRT1, BCOR, and BCL2L11, while acute resolving infection was marked by the higher expression of regulators of T cell differentiation and memory, e.g., TCF7 and its transcriptional target LEF1 [22]. This suggests that CD8^+^ T cells T cell differentiation might be negatively impacted very early during chronic infection, manifested as the rapid dysregulation of genes that are crucial for orchestration and direction of adaptive T cell responses.

## 3. Reversibility of T Cell Exhaustion in HCV Infection

Understanding the molecular pathways of T cell exhaustion, in particular the contribution of inhibitory receptors, has helped to identify potential strategies aiming to restore T cell functionality and to improve infection control [66,67]. These included the specific blocking of immunosuppressive cytokines, iRs or their ligands with monoclonal antibodies, antiviral treatment and vaccination (Figure 1). Promising results obtained by blocking of iRs signaling pathways in HIV and LCMV infections have prompted similar studies in in vitro models of HCV infection (Table 1). Because of their better characterization, most studies explored the aspect of CD8^+^ T cells’ exhaustion.

Blockades of PD-1 and/or Tim-3 pathways using monoclonal antibodies could restore functional features of HCV-specific CD8^+^ T cells, such as proliferation and effector cytokines secretion, whereas improvement in the cytolytic function was observed exclusively in the case of Tim-3 blockade [18,68,69,70,71]. These findings are congruent with the concept of hierarchical model of functional T cell exhaustion, with an easier restoration of proliferation capacity, followed by the restoration of effector cytokines production, but with a general lack of effect on cytotoxicity.

Encouraging beneficial effects of in vitro experiments warranted studies employing in vivo immunotherapy of chronic HCV infection using antibodies against iRs (Table 1) [72,73,74]. Fuller et al. [72] investigated the effect of anti-PD-1 antibodies treatment on T cell responses in three chimpanzees with chronic HCV infection. A significant reduction in HCV viral load without signs of hepatocellular injury was observed in one animal, which rebounded when antibody treatment was discontinued. The viral load drop in this animal was accompanied by the restoration of intrahepatic CD4^+^ and CD8^+^ T cell immunity to multiple HCV proteins. 

Gardiner et al. [73] presented the findings of a proof-of-concept, placebo-controlled, single-ascending-dose study in 54 patients with chronic HCV infection treated with PD-1-targeting nivolumab (*BMS*-*936558*). Five patients were reported to respond with reduction in viral load, including two patients with HCV RNA levels below the lower limit of detection. Importantly, treatment was not related to any evidence of immune deficit, since neither clinically relevant changes in immunoglobulin subsets nor changes in mean serum levels of major cytokines were observed during follow-up. Furthermore, substantial quantitative changes in immune cell subsets were not evident and cell counts generally returned to pretreatment levels after one week of treatment.

A similar pilot clinical trial was conducted by Sangro et al. [74] to test the antitumor and antiviral effect of tremelimumab (anti-CTLA-4 monoclonal antibody) in 20 patients with chronic HCV infection and HCC. Tremelimumab induced a decrease in viral load in most patients followed-up for at least three months. Fifteen percent of patients experienced a transient complete viral response. In parallel, an increased number of virus-specific, IFN-γ-producing lymphocytes were detected, suggesting that antiviral effect was most likely a result of enhanced T-cell-mediated immunosurveillance. *Quasispecies* changes, manifested as new emerging hypervariable region 1 variants, were observed, coinciding with the second cycle of tremelimumab. 

The use of immune checkpoint inhibitors in cancer immunotherapy was shown to increase the risk of immune-related adverse events (irAEs) as a result of the elevated activity of immune cells [75]. IrAEs can affect multiple organs, i.e., liver, skin, digestive system, lung, endocrine glands and potentially other tissues [75]. In particular, liver damage manifesting as hepatitis occurs in 1–17% of treated patients [76]. Nevertheless, in the study of Sangro et al., a good safety profile was recorded, including no severe irAEs requiring the administration of steroids.

Cytokines are also regarded as an attractive therapeutic target for the modulation of immune response in chronic viral infections. It was shown that selective blockade of the IL-10 receptor (IL-10R) can result in virus control and the reinvigoration of immune response in chronic LCMV infection in vivo [77]. Translating these findings into HCV infection, Rigopoulou et al. [78] showed that the in vitro monoclonal antibody-induced blockade of IL-10R resulted in a dose-dependent increase in CD4^+^ T cell proliferative responses to the HCV core, as well as non-structural proteins 3 (NS3) and 4 (NS4). Furthermore, the blockade of IL-10R altered the balance in favor of type 1 antiviral T cell immunity with an increased frequency of HCV-specific, IFN-γ-producing CD4^+^ T cells.

Taken together, the available findings, although sparse, imply that HCV-induced immune dysfunction may be reversible. They demonstrated that the restoration of proliferation and effector functions of HCV-specific T cells could be observed in vitro and in vivo, the latter being accompanied by durable objective responses and good safety profiles. However, because of the recent advances in and introduction of highly effective therapies based on direct-acting antivirals (DAA), the scope of T cell research has shifted, and the abovementioned immunotherapeutic approaches have been mostly abandoned.

## 4. Impact of DAA Treatment of Chronic HCV Infection on T Cell Exhaustion

In recent years, substantial progress has been achieved in treatment of chronic HCV infection, which is expected to reduce the extent of virus-related morbidity and mortality (reviewed in [79]). The assessment of therapy effectiveness involves the periodic screening of HCV RNA in patient’s serum by polymerase chain reaction (PCR). A sustained virologic response (SVR), defined as the absence of HCV RNA evaluated at 12 weeks (formerly 24 weeks) post-treatment, is regarded as a therapeutic success [80]. Until recently, the standard of care for chronic hepatitis C was a combination of pegylated-IFN-α (PEG-IFN-α) with ribavirin (RBV). IFN-α plays an immunoregulatory, antiviral and anti-proliferative role, while ribavirin inhibits viral RNA polymerase [80]. The newly recommended IFN-free treatment schemes include DAA drugs (NS3/4A, NS5A and NS5B inhibitors) [81]. DAAs exhibit antiviral activity via interference with HCV replication cycle (inhibition of HCV polyprotein maturation and HCV RNA synthesis) [82]. These treatment regimens are short (i.e., last routinely 8–12 weeks), safe, well tolerated, highly effective (SVR rates above 95%), and can be optimized by combining drugs with synergistic or additive effects [83]. 

The impact of anti-HCV treatment commenced in chronic phase of infection on already-established T cell exhaustion is largely unknown and commonly inconclusive, which may be due to methodological differences (e.g., assessing various immunological effects of treatment, small patient cohorts, different treatment schemes, inclusion of patients infected with different HCV genotypes as well as different follow-up time points) [70,84,85,86,87]. Theoretically, the successful therapy of chronic HCV infection might reverse the functional T cell exhaustion by a number of mechanisms, e.g., by a rapid reduction in viral load and immune system stimulation with viral antigens or the elimination of viral proteins known to inhibit immune responses [15,88,89,90].

Studies of T cell function in IFN-based therapy of chronic HCV infection have shown that it reduces the numbers and impairs the functional potential of antiviral T cells [65,91,92,93]. While successful IFN-based treatment of acute HCV infection results in highly functional T cell responses comparable to those in spontaneous resolvers, SVR attained by IFN-based treatment of chronic HCV infection does not result in the functional restoration of HCV-specific CD8^+^ T cells [92,94]. This implies that, in chronic HCV infection, the endogenous T cell population does not contribute to the IFN-based treatment’s success [65,91,92,93]. In contrast, the restoration of antiviral immunity, manifested by the reversal of the exhausted T cells’ phenotype and immune-driven elimination of residual viral replication, may be necessary for successful DAA treatment, since the presence of HCV-RNA in serum at the end of treatment (EOT) does not preclude SVR [95,96]. In agreement with that, some findings suggest that DAA treatment may result in certain positive effects on the T cell immune function or phenotype (Table 2 and Table 3).

Some studies have demonstrated that DAA treatment of chronic HCV infection leads to the reconstitution of peripheral T cell populations, as evidenced by an increase in the frequency of CD4^+^ [98,100] and CD8^+^ T cells [100], and shift toward T_em_ (effector memory) population [97,98], with a concomitant decrease in the naïve T cell subset [97]. Effector function reinvigoration was also observed, manifested by increased frequencies of circulating T helper and cytotoxic T cells, producing IFN-γ, IL-17, and IL-22 [99]. Furthermore, a reduction in the expression of PD-1 [97] as well as TIGIT [97,98] on both CD4^+^ and CD8^+^ T cells was reported. Similarly in our study, DAA-treatment resulted in significant decreases in CD4^+^PD-1^+^Tim-3^+^ and CD8^+^PD-1^+^Tim-3^+^ T cell frequencies to levels observed in controls, while CD8^+^PD-1^+^ T cells significantly increased (manuscript submitted). Furthermore, the DAA effect was also observed in decreased IL-10 plasma levels.

Most studies focused on the impact of successful DAA treatment of chronic HCV infection on the status of CD4^+^ and CD8^+^ T cell activation. While one study did not show any significant changes after treatment [99], other studies did show a decline in T cell activation status, as evidenced by reduced HLA-DR and/or CD38 expression after treatment [97,100,101,102]. This was also demonstrated in the case of HIV-1/HCV co-infection [102]. However, Vranjkovic et al. [103] observed that the impact of successful DAA treatment on T cell activation depends mostly on the status of liver fibrosis. The hyperfunctional activity of peripheral CD8^+^ T cell subsets (naïve, effector, early effector memory, late effector memory and central memory) was sustained in HCV-infected patients with liver fibrosis (F4) up to a year after treatment, particularly manifesting as elevated perforin production and cellular cytotoxicity when compared to patients with minimal fibrosis (F0-1). Furthermore, DAA treatment had no effect on elevated concentrations of systemic inflammatory cytokines and decreased levels of inhibitory TGF-β in plasma of F4 patients, suggesting that HCV infection and advanced liver disease result in a long-lasting immune activating microenvironment. According to the authors, the sustained hyperfunction of CD8^+^ T cells long after DAA treatment may have significant long-term consequences, including compromised antitumor immunity, often aggressive forms of HCC, increased risk of HCC recurrence [111] and extrahepatic cancers [112]. Other concerns include potential failure to generate effective HCV vaccine responses and vulnerability to HCV re-infection [113].

The abovementioned studies mostly concerned the effect of DAA-based treatment on the total peripheral T cell populations, typically not investigating HCV-specific T cells. Because of limited numbers of such cells in circulation, only a few studies addressed this issue [97,98,104,106,107] (Table 3). 

Romani et al. [104] found the SVR-predictive value of PD-1^+^ HCV-specific CD8^+^ T cell subset with cytotoxic capacity (degranulation and cytokine production) in short-duration DAA treatment. Higher levels of PD-1^+^ CD8^+^ HCV-specific T cells were observed in patients who achieved SVR, both at baseline and at EOT, compared to relapsers, which points to the essential role of these cells in DAA-mediated viral clearance. However, 12 weeks after EOT, the frequency of this subset of cells was significantly reduced exclusively in the SVR^+^ group, which possibly reflects the elimination of PD-1^+^ HCV-specific cells after successful antigen removal. Thus, these results confirm the hypothesis of the active role of immunity in DAA-mediated viral clearance. 

Burchill et al. [98] observed that although the frequency of PD-1^+^ HCV-specific T cells significantly decreased, the frequency of HCV-specific CD8^+^ T cells was not significantly altered post-successful DAA treatment.

Martin et al. [105] showed that some functional capabilities, e.g., proliferation, could be renewed. A significant increase in the frequency of HCV-specific CD8^+^ T cells after in vitro expansion was observed in most of SVR^+^ patients from baseline to 24 weeks after completion of treatment, but not in patients with treatment failure. 

Shrivastava et al. [97] demonstrated that successful DAA treatment of HCV infection in HIV-1/HCV co-infected patients led to the improvement in HCV-specific T cell function, including cytokine production (IL-2 and IFN-γ), polyfunctionality (measured as an increase in the proportion of cells co-expressing IFN-γ and TNF-α) and cytolytic capacity (increase in CD107A expression and perforin and granzyme B secretion). Comparing therapies with two or three different DAAs, they showed that the most profound restoration of HCV-specific immune responses was observed in the group of patients treated with a regimen that inhibits three distinct stages of the HCV life cycle. Whether this was due to more potent suppression of HCV in vivo or an independent effect on the immune system remained unknown.

Wieland et al. [106] showed that the restoration of HCV-specific CD8^+^ T cells’ ability to proliferate was associated with changes in their composition. While terminally exhausted HCV-specific CD8^+^ TCF-1^-^CD127^-^PD1^hi^ T cells disappeared after antigen elimination, the memory-like HCV-specific CD8^+^ T cells (TCF-1^+^CD127^+^PD-1^+^), with a retained ability of self-renewal and proliferation, persisted, even in the absence of HCV antigen. Moreover, these cells displayed the capacity of robust secondary expansion in a patient with a viral relapse. However, they did not fully resemble memory HCV-specific CD8^+^ T cells observed after spontaneous viral clearance since they displayed higher PD-1 and Eomes expression, indicative of T cell exhaustion and impaired cytokines production.

Han et al. [107] observed that although successful DAA treatment increased the proliferative capacity of HCV-specific CD8^+^ T cells, their ex vivo function, which manifested as IFN-ϒ production capabilities, cytotoxicity as well as diminished exhausted marker expression, was only transient (i.e., observed only at week 4 of treatment, which coincided with viral clearance), but attenuated and returned to baseline levels at SVR12. While TCF-1^+^CD127^+^PD-1^+^ HCV-specific CD8^+^ T cells responsible for recall proliferation after antigen re-challenge remained unchanged over time, ex vivo HCV-specific CD8^+^ T cell frequency decreased at SVR12, including antigen-experienced (KLRG1^+^CCR7^−^) HCV-specific CD8^+^ T cell subset. In contrast, patients experiencing viral breakthrough or relapse exhibited defective restoration of HCV-specific T cell immunity, manifested as significantly lower T cell responses at any timepoint of observation. These findings suggest that the ex vivo function of HCV-specific T cells from chronically infected patients may not be enhanced after successful DAA treatment, despite transient functional restoration during early treatment.

Aregay et al. [108] observed that HCV clearance following DAA therapy was not able to fully restore HCV-specific CD8^+^ T cells function. The expression of exhaustion markers (PD-1, Tim-3, LAG-3 and CD5) on HCV-specific CD8^+^ T cells, impaired cytokine production (IFN-ϒ, MIP-1β), mitochondrial dysfunction and metabolic deregulation (reduced mitochondrial polarization, increased mitochondrial mass and increased mitochondrial ROS level), and did not alter after successful DAA treatment. Furthermore, the impaired proliferative potential of HCV-specific CD8^+^ T cells was only partially restored. In contrast, a significant reduction in HCV-specific CD8^+^ T cells expressing activation markers (CD38 and HLA-DR), which could be due to the associated loss of terminally differentiated CD39^+^ HCV-specific CD8^+^ T cells, was seen following HCV elimination. Similar to Wieland et al.’s and Han et al.’s studies [106,107], memory-like HCV-specific CD8^+^ T cells (TCF-1^+^CD127^+^PD-1^+^) remained unaltered after HCV clearance. Interestingly, the proliferative capacity could be increased upon PD-1/PD-L1 pathway blockade exclusively in HCV-specific CD8^+^ T cells, whose proliferative potential was not restored after HCV clearance by means of DAA therapy. 

Although epidemiological data show cases of HCV re-infection after DAA-mediated viral clearance, it is currently unclear how the eventual restoration of CD8^+^ HCV-specific T cell response would assure protection or influence the clinical course of re-infection [114]. However, a recent study employing a chimpanzee model of chronic HCV infection has demonstrated that the HCV-specific CD8^+^ T cell population which persisted after DAA-mediated viral clearance did not prevent the development of chronic infection after re-infection [115]. 

In contrast to CD8^+^ T cells, there are almost no data on DAA-treatment-induced alterations within the CD4^+^ HCV-specific T cell subset. In the study by Hartnell et al. [109], it was demonstrated that helper responses in chronic HCV infection was infrequently detected and poorly functional, and did not consistently recover following HCV cure, since DAA treatment did not promote proliferative capacity and TNF-α, IFN-ϒ and MIP-1β production by HCV-specific CD4^+^ T cells.

In contrast, Smits et al. [110] observed that the very low baseline frequency of HCV-specific CD4^+^ T cells increased within the initial two weeks of DAA treatment. Although percentages of HCV-specific CD4^+^ T cells expressing PD-1, BTLA, and TIGIT were maintained during observation (from baseline to follow-up (SVR24)), analyses of the mean fluorescence intensity (MFI) revealed a significant reduction in the expression levels of PD-1. Importantly, although cells with a Th1 phenotype were the predominant subset at baseline, cells with phenotypic and transcriptional characteristics of follicular T helper cells (Tfh) significantly increased from baseline to follow-up (SVR24), suggesting the antigen-independent survival of this subset. These changes were accompanied by a decline in the germinal center activity and HCV-specific neutralizing antibodies, indicating that these cells may be involved in maintaining HCV-specific humoral immunity during chronic infection.

CD4^+^ regulatory T cells (Treg, Foxp3^+^ CD25^+^ CD4^+^) limit the in vitro responses of effector CD8^+^ T cells via suppression of their activation [116]. Although the direct role of Tregs in exhaustion of CD8^+^ T cells remains unclear, they may play a role, considering that Tregs are a source of immune-suppressive IL-10, TGF-β and IL-35 [66]. Langhans et al. [117] showed that the percentage of Tregs was significantly higher in chronically HCV-infected patients than in healthy controls. After successful DAA therapy, their frequency slightly decreased, although up to 51 ± 14 weeks post-EOT it did not reach levels similar to those observed in healthy controls. The expression of activation/regulation markers on Tregs (i.e., GARP, OX-40, CTLA-4, GITR, Tim-3 and galectin-9) was only slightly reduced after therapy and remained higher than in healthy controls. Wu et al. [118] demonstrated that, in patients treated with DAAs for chronic HCV infection, the frequency of Tregs and their inhibitory function on proliferation of CD4^+^CD25^−^ effector T cells decreased from baseline to EOT, although it subsequently increased from EOT to SVR12 to levels close to baseline. Consequently, these changes coincided with fold changes in IFN-γ and TNF-α secretion, which were the highest at EOT and then decreased at SVR12, whereas an inverse relationship was observed in the case of IL-10, which decreased at EOT and then increased at SVR12. Thus, the concept of decreases in the frequency and activation status of Tregs toward physiological levels following DAA treatment did not find support in the above studies. 

## 5. Effect of Vaccines on T Cell Exhaustion in Chronic HCV Infection

Although the advent of DAAs has revolutionized the management of chronic HCV infection, in many countries the scale of new HCV cases outnumbers those that are cured [119]. In addition, there is still a risk of re-infection in patients after successful anti-HCV treatment [114,120]. Therefore, HCV vaccine development is paramount to limit the spread of the virus. The main goal of HCV vaccination is to increase the antiviral activity of virus-specific CD4^+^ and CD8^+^ T cells to protect against HCV infection or re-infection. Hartnell et al. [109] demonstrated that the immunization of healthy volunteers with a novel experimental heterologous recombinant vaccine (chimpanzee adenovirus (ChAd3) combined with modified vaccinia Ankara virus (MVA), encoding the non-structural region of HCV (NSmut)) in a prime/boost regimen, induced the generation of long-lived memory CD127^+^ HCV-specific CD4^+^ T cells with increased expression of CD28 co-stimulatory receptor, while the expression of T-bet decreased. Furthermore, vaccination promoted the vigorous production of TNF-α, IFN-γ and MIP-1β, as well as the robust proliferative capacity of HCV-specific CD4^+^ T cells. Similar qualities of response were demonstrated in patients after spontaneous virus elimination, however, as mentioned above, this was not observed in patients after successful DAA treatment of chronic infection [109]. 

Multi-specific, high-magnitude responses observed following HCV vaccination in healthy volunteers have prompted an investigation as to whether therapeutic vaccination of patients with chronic HCV infection would result in the generation of vigorous immunity. However, Swadling et al. [121] and Kelly et al. [122] demonstrated no significant induction of HCV-specific T cells and no impact on viral load after ChAd3-NSmut/MVA-NSmut prime/boost vaccination of chronically infected patients treated with PEG-IFNα/rib (with lower baseline viral load) or untreated (with higher baseline viral load). In both studies, vaccine-induced T cells were functionally impaired and did not proliferate after stimulation. Strikingly, the expression of exhaustion markers (PD-1, Tim-3, CTLA-4 and 2B4) on HCV-specific T cells, as well as the transcription factor T-bet/Eomes co-expression pattern, were similar to those seen in healthy volunteers long-term after vaccination [121]. Interestingly, T cell induction was the lowest in patients in whom vaccine immunogen and circulating virus variants displayed the same sequence, which may have been a result of T cell exhaustion. Conversely, T cells were induced in the context of sequence mismatch between vaccine immunogen and circulating virus, despite a common failure to recognize circulating epitope variants and only partially functional phenotype [121,122]. 

The effect of HCV vaccination on antiviral CD8^+^ T-cell function has also been studied in two chronically infected chimpanzees during treatment with DAAs. Callendret et al. showed that although a combination of DAAs and genetic vaccines encoding the HCV NSmut improved peripheral blood CD8^+^ T-cell function, vaccine-induced antiviral CD8^+^ T-cells mostly did not recognize circulating persistent virus [123]. These cells did not preclude the replication of DAA-resistant HCV variants which evolved during therapy. Furthermore, exhausted intrahepatic CD8^+^ T-cells targeting conserved epitopes did not expand after vaccination, and failure to control HCV replication was likely caused by compartmentalized CD8^+^ T-cell exhaustion in the liver, as manifested by high PD-1 expression. These observations point out that the major challenge in successful therapeutic vaccine design would be to overcome T cell exhaustion in chronic HCV infection.

## 6. Conclusions and Future Perspectives

The long-term consequences of the impairment of T cell responses in chronic HCV infection are not entirely characterized, though they may be essential in the context of clinical course of infection, re-infection or treatment-mediated viral clearance and vaccine design. Furthermore, it is still uncertain whether a complete restoration of HCV-specific T cell response may be feasible, and whether this restoration would guarantee protection or influence the clinical course of re-infection.

The available studies imply that chronic HCV infection is accompanied by substantial and progressive phenotypic and functional alterations, including upregulation of multiple inhibitory receptors, which downregulate the functional and proliferative potential of the responding cells. Attempts to reverse the effects of compromised immune response quality included specific blockades of inhibitory receptors or immunosuppressive cytokines by means of monoclonal antibodies. In most studies, a restoration of functional competence (proliferation and effector functions) of HCV-specific T cells could be observed in vitro and in vivo, the latter accompanied by viral load decline. This implies that HCV-induced immune dysfunction may be reversible. Another investigated aspect was whether the elimination of viral antigens during chronic infection by means of HCV-targeted direct acting antiviral treatment could restore T cell function. Although the collected data were largely fragmentary and sometimes even contradictory, in most of them there is evidence that DAA treatment may result in at least partial restoration of T cell immune function. Absence of these hallmarks of immune restoration in non-responders is indicative of an active role of immunity in DAA-mediated viral clearance. Despite initial enthusiasm, they also suggest that a complete restoration, comparable to that seen after spontaneous viral clearance, may not be attained, which points out that long-term antigenic stimulation imprints an irreversible change on the T cell compartment. Understanding the mechanisms of HCV-induced immune dysfunction and barriers to immune restoration following viral clearance is of the utmost importance to diminish the possible long-term consequences of chronic HCV infection. 

## Figures and Tables

**Figure 1 viruses-12-00799-f001:**
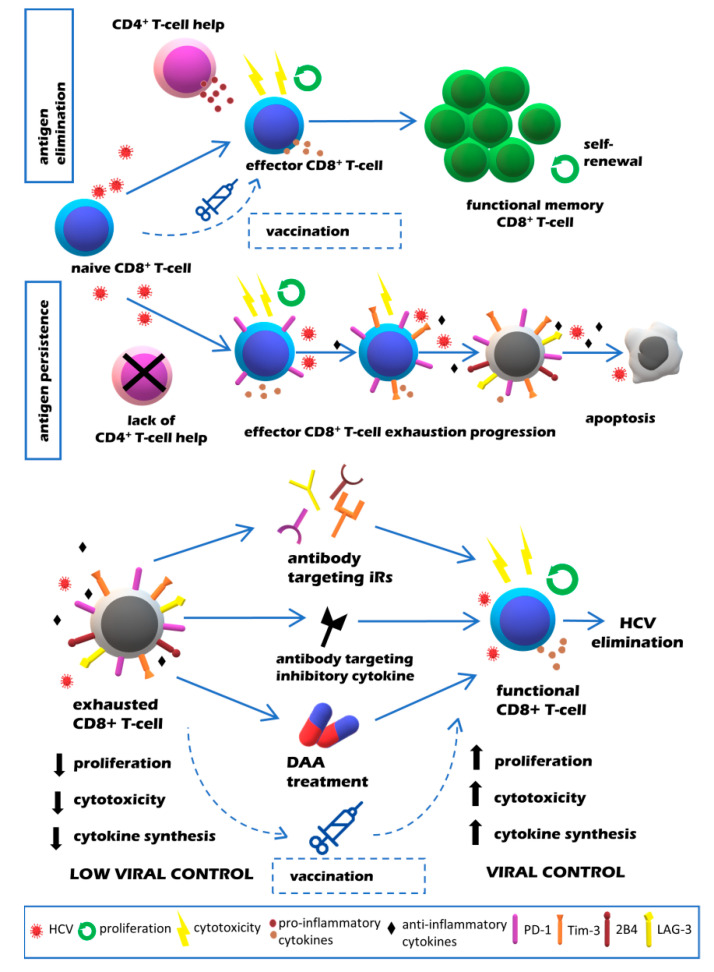
Quality of CD8^+^ T cell responses in resolving vs. chronic HCV infection (upper panel) and potential strategies aimed at restoration of exhausted T cell function in chronic HCV infection (lower panel).

**Table 1 viruses-12-00799-t001:** Studies reporting the effect of immune checkpoint inhibitor blockades on CD8+ T cell function in HCV infection.

Reference	Immune Checkpoint Blocked	Stage of HCV Infection	Number of Subjects	Character of the Study	Results
Golden Mason et al. [68]	PD-ligand 1 (PD-L1) PD-ligand 2 (PD-L2)	chronic	7	In vitro	↑ proliferation of HCV-specific CD8^+^ T cells ↑ IFN-ϒ and IL-2 secretion by HCV-specific CD8^+^T cells
Golden Mason et al. [69]	Tim-3	chronic	4	In vitro	↑ proliferation of HCV-specific CD8^+^ T cells ↑ IFN-ϒ secretion by HCV-specific CD8^+^ T cells ↓ IL-10 secretion by HCV-specific CD8^+^ T cells
Penna et al. [18]	PD-L1	chronic	8	In vitro	↑ expansion of HCV-specific CD8^+^ T cells ↑ frequency of both IFN-γ– and IL-2–secreting HCV-specific CD8^+^ T cells
Urbani et al. [70]	PD-L1	acute	8	In vitro	↑ expansion and IFN-γ and IL-2 production but not the cytolytic activity of HCV-specific CD8^+^ T cells.
McMahan et al. [71]	Tim-3, PD-L1, PD-L2	acute/chronic	6/4	In vitro	↑ proliferation of HCV-specific CD8^+^ T cells achieved by either PD-1 or Tim-3 blockade ↑ cytotoxicity of HCV-specific CD8^+^ T cells (increased expression of CD107a, killing of hepatocytes cell line expressing cognate HCV epitopes) achieved exclusively by Tim-3 blockade
Fuller et al. [72]	PD-1	chronic	3 chimpanzees	In vivo	↓ HCV viral load in one of three treated animals ↑ frequencies and IFN-ϒ production of intrahepatic HCV-specific CD4^+^ and CD8^+^ T cells in the same animal
Gardiner et al. [73]	PD-1	chronic	54	In vivo	↓ viral load in five patients (two patients achieved undetectable HCV RNA)
Sangro et al. [74]	CTLA-4	chronic	20	In vivo	↓ viral load sustained in most patients for 3 months follow-up; transient complete viral response in 15% of patients during follow-up ↑ HCV-specific T cell response (IFN-ϒ production)

↑ —increase; ↓ —decrease.

**Table 2 viruses-12-00799-t002:** Studies reporting the effect of direct-acting antiviral (DAA) treatment on peripheral T cell phenotype or function in chronic HCV infection.

Reference	HCV Genotype	Number of Subjects	Effect of Treatment/Effect of Successful Treatment	Follow-up	Results
Shrivastava et al. [97]	1	22 HIV/HCV co-infected	Effect of successful treatment	12 weeks after the end of treatment (EOT) (sustained virologic response (SVR) 12)	↓ PD1 and TIGIT expression on CD4^+^ and CD8^+^ T cells ↓ Eomes^hi^ T-bet^lo^ CD4^+^ and CD8^+^ T cells ↑ T-bet^hi^ Eomes^lo^ CD4^+^ and CD8^+^ T cells ↓ BLIMP-1 expression on CD4^+^ T cells ↓ CD38 expression on both CD4^+^ and CD8^+^ T cells ↑ T_em_ (effector memory) population ↓ naïve T cell subset
Burchill et al. [98]	1a/1b	19	Effect of successful treatment	24 weeks post-EOT (SVR24)	↑ frequency of CD4^+^ T cells; ↓ expression of TIGIT on CD4^+^ and CD8^+^ T cells; ↑ percentage of T_em_ in both CD4^+^ and CD8^+^ T cells compartments
Najafi Fard et al. [99]	1–4	HCV mono-infection *n* = 18; HCV/HIV-1 co-infection (*n* = 17)	Effect of successful treatment	12 weeks post-EOT (SVR12)	↑ peripheral CD4^+^ and CD8^+^ T cells producing IFN-γ, IL-17, and IL-22 no significant impact on the status of CD4^+^ and CD8^+^ T cells activation
Meissner et al. [100]	1	95	Effect of treatment	up to 20 weeks after treatment initiation	↑ peripheral CD4^+^ and CD8^+^ T cells early after treatment initiation ↓ HLA-DR^+^CD38^+^ T-cells during observation ↑ expression CXCR3 on T cells early after treatment initiation
Lattanzi et al. [101]	1–4	45	Effect of treatment	at first month of treatment (T1), at EOT (T2) and 12 weeks post-EOT (T3, SVR12)	stable percentage of CD4^+^ and CD8^+^ T cells at T1 when compared to baseline ↓ HLA-DR^+^ CD38^+^ CD4^+^ and CD8^+^ T cells at T3 with respect to baseline
Emmanuel et al. [102]	NA	HCV mono-infection (*n* = 161); HIV/HCV co-infection (*n* = 59)	Effect of successful treatment	1 or 2 years post-SVR	↓ HLA-DR^+^CD38^+^ CD4^+^ and CD8^+^ T cells in both HCV infection and HIV/HCV co-infection
Vranjkovic et al. [103]	NA	18	Effect of successful treatment	24 weeks post-SVR12	phenotypic distribution of peripheral CD8^+^ T cell subsets in patients with advanced liver fibrosis (F4) different from those with minimal fibrosis (F0-1) which remained unchanged after viral elimination sustained hyperfunctional activity (perforin production and cytotoxicity) of CD8^+^ T cell subsets in patients with liver fibrosis (F4) up to a year post-treatment initiation sustained elevated concentrations of systemic inflammatory cytokines and decreased levels of TGF-β in plasma of patients with liver fibrosis (F4)

↑—increase; ↓—decrease; NA—not available.

**Table 3 viruses-12-00799-t003:** Studies reporting the effect of DAA-treatment on peripheral HCV-specific T cell phenotype or function in chronic HCV infection.

Reference	HCV Genotype	Number of Subjects	Effect of Treatment/Effect of Successful/Unsuccessful Treatment	Follow-up	Results
Romani et al. [104]	1a/1b	26	Effect of successful/unsuccessful treatment	at the end of treatment (EOT), at week 4 and 12 weeks post-EOT (sustained virologic response (SVR) 12)	higher levels of PD-1^+^ HCV-specific T cells at baseline and at EOT in patients who achieved SVR ↓ PD-1^+^ HCV-specific T cell subset at SVR12 in responders
Burchill et al. [98]	1a/1b	7	Effect of successful treatment	24 weeks post -EOT (SVR24)	no significant change in the frequency of HCV-specific CD8^+^ T cells ↓ PD-1 expression
Martin et al. [105]	1	51	Effect of successful/unsuccessful treatment	treatment week 4, 12 and 24 weeks post-treatment (SVR24)	↑ HCV-specific CD8^+^ T cells frequency after in vitro expansion in patients with SVR from baseline to 24 weeks after completion of treatment no change in HCV-specific CD8^+^ T cells frequency in patients with treatment failure
Shrivastava et al. [97]	1	22 HIV-1/HCV co-infected	Effect of successful treatment	12 weeks post-EOT (SVR12)	↑ HCV-specific CD8^+^ T cells ↑ IL-2 and IFN-γ production ↑ polyfunctionality (co-expression of IFN-γ and TNF-α) ↑ cytolytic capacity (CD107A expression and perforin and granzyme B secretion)
Wieland et al. [106]	1a/1b	21	Effect of successful treatment	at EOT and 12 weeks post-EOT (SVR12)	↓ terminally exhausted HCV-specific CD8^+^ T cells (TCF-1^-^CD127^-^PD1^hi^) after antigen elimination persistence of memory-like HCV-specific CD8^+^ T cells (TCF-1^+^CD127^+^PD-1^+^) with ability of self-renewal and proliferation
Han et al. [107]	1b/2a	41	Effect of successful/unsuccessful treatment	treatment week 4, 12, 24 (EOT) and 12 weeks post-treatment (SVR12) or week 4, 12 (EOT), and 12 weeks post-treatment (SVR12)	↑ HCV-specific CD8^+^ T cell response (IFN-ϒ production, cytotoxicity) at week 4, which diminished at later weeks ↓ PD-1^+^Eomes^hi^T-bet^low^ HCV-specific CD8^+^ T cells at week 4 ↓ HCV-specific CD8^+^ T cell frequency at SVR12, including antigen-experienced KLRG1^+^CCR7^−^ HCV-specific T cells no change in TCF-1^+^CD127^+^PD-1^+^ HCV-specific CD8^+^ T cells responsible for recall proliferation over observation time defective restoration of HCV-specific T cell responses in SVR^-^ group
Aregay et al. [108]	1a/1b	40	Effect of successful treatment	at EOT and 24 weeks post-EOT (SVR24)	unaltered expression of PD-1, Tim-3, LAG-3 and CD5 on HCV-specific CD8^+^ T cells sustained impaired IFN-ϒ, MIP-1β production, mitochondrial dysfunction and metabolic deregulation ↓ HLA-DR^+^CD38^+^ HCV-specific CD8^+^ T cells maintenance of memory-like TCF-1^+^CD127^+^PD-1^+^ HCV-specific CD8^+^ T cells
Hartnell et al. [109]	NA	21	Effect of successful treatment	average 6 weeks post-treatment (range 0–26 weeks)	unchanged proliferative capacity and cytokine production (TNF-α, IFN-ϒ MIP-1β) of exhausted HCV-specific CD4^+^ T cells
Smits et al. [110]	1, 2, 3	40	Effect of successful treatment	week 2, either 8, 12, 16 or 24 week of treatment (EOT) and 24 weeks post-treatment (SVR24)	↑ HCV-specific CD4^+^ T cells within the initial two weeks of treatment unchanged percentages of HCV-specific CD4^+^ T cells expressing PD-1, BTLA and TIGIT ↑ follicular T helper cells (Tfh) ↓ germinal center activity and HCV-specific neutralizing antibodies

↑—increase; ↓—decrease; NA—not available.

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
