# Peer review of "Reversal of T Cell Exhaustion in Chronic HCV Infection"

_viruses, 2020, doi:10.3390/v12080799_

Round 1
Reviewer 1 Report
The authors made all the necessary adjustments to the text, answered all my questions.
Author Response
Thank you
Reviewer 2 Report
The authors have addressed the reviewers' comments in a satisfactory manner. In my opinion the review is suitable for publishing in its current form, with minor language revisions after the changes that were applied.
Author Response
- The Reviewer commented that the manuscript requires minor language revisions. In response, this has been done (please see red-marked changes).
Reviewer 3 Report
In the revised version of the manuscript the authors have addressed my points of critique in a satisfactory manner.
Minor comments
line 149: replace the word "animals" with "chimpanzees"
line 153: Ref 64 does not seem to be the correct reference here
Author Response
- The Reviewer asked to replace the word "animals" with "chimpanzees" (former page 4 line 149). This has been done (now page 4 line 148).
- The Reviewer pointed out that ref #64 was incorrectly used (former page 4 line 153). Authors do apologize for this mistake. In response, it was replaced by a suitable reference (#16 in the revised manuscript, now page 4 line 152).
This manuscript is a resubmission of an earlier submission. The following is a list of the peer review reports and author responses from that submission.
Round 1
Reviewer 1 Report
Hepatitis C virus (HCV)-specific T cells are functionally impaired in chronic hepatitis C (CHC). Even though HCV can now be rapidly and sustainably cleared from chronically infected patients, the repercussions of HCV clearance on virus-specific T cells remain elusive. The Review of Sylwia Osuch et al. is devoted to the important problem: reversal of T cell exhaustion in chronic HCV infection. Part of the review concerns of studies reporting the effect of immune checkpoint inhibitor blockades on CD8+ T cell function in HCV infection. Most of the review describes the studies reporting the effect of DAA‐treatment on peripheral HCV‐specific T cell phenotype or function in chronic HCV infection. The review is logically written, there are the necessary tables with data and one illustration.
Main remarks
Add information about possible complications and liver damage related to immune checkpoint inhibitors.
Add information about the effect of DAA‐treatment on peripheral HCV‐specific T cell (for example, PMID: 31295532, etc.).
Pay more attention to the effect of vaccines on the T-cell response in CHC patients (for example, PMID: 32012325, PMID: 27490575, PMID: 26474390, etc.).
Discuss the role of CD4 + Regulatory T cells that persist after successful DAA-treatment (for example, PMID: 28040549, PMID: 31559553, etc.).
Page 5, Table 1 - column Results - edit “In 6 of 8 cases… “, the sentence is repeated 3 times.
Reviewer 2 Report
Kindly find attached a detailed reviewing report.

Reviewer 3 Report
In the review entitled “Reversal of T cell Exhaustion in Chronic HCV Infection” Osuch et al. summarize and review research on T cell exhaustion and its potential reversion during HCV infection. In general, the topic of the review is timely and highly relevant. In particular chapter 5 and table 2-3 of the manuscript comprehensively summarize relevant current data.
However, other parts of the manuscript (chapter 2 and 4) do not discuss the most recent advances in the field and lack current citations. Instead, a significant proportion of these chapters is dedicated to the description of rather outdated research. This limits the usefulness of this manuscript in providing a current overview of research in the field of T cell exhaustion during HCV infection.
Specific points of concern:
Chapter 2: this is not a comprehensive overview of the current research on T cell responses and T cell exhaustion during HCV infection. Many relevant citations are missing. Current advances, such as insights into the transcriptomic profile of HCV-specific CD4+ and CD8+ T cells during acute resolving versus chronic infection are not summarized. New concepts of T cell exhaustion, such as the role of the transcription factors TCF-1 and TOX in regulating T cell exhaustion in different T cell subsets, are not discussed. The authors should review the current literature and update this chapter accordingly.
Chapter 4: this chapter is rather long and dedicated to the summary of mostly outdated research. In table 1 the newest reference is 7 years old. Checkpoint inhibitor blockade for the reversal of T cell exhaustion during HCV infection is not an active research topic anymore. This fact is acknowledged by the authors in lines 70-71 of the manuscript. Thus, I suggest the authors significantly shorten chapter 4 and instead extend chapter 2 to provide a more current view of the research ongoing in the field.